# Ovule Development and *in Planta* Transformation of *Paphiopedilum* Maudiae by *Agrobacterium*-Mediated Ovary-Injection

**DOI:** 10.3390/ijms22010084

**Published:** 2020-12-23

**Authors:** Bai-Xue Luo, Li Zhang, Feng Zheng, Kun-Lin Wu, Lin Li, Xin-Hua Zhang, Guo-Hua Ma, Jaime A. Teixeira da Silva, Lin Fang, Song-Jun Zeng

**Affiliations:** 1Key Laboratory of South China Agricultural Plant Molecular Analysis and Gene Improvement, South China Botanical Garden, Chinese Academy of Sciences, Guangzhou 510650, China; xiangshangdexue@163.com (B.-X.L.); zhangli112113@163.com (L.Z.); zhengfeng@scbg.ac.cn (F.Z.); wkl8@scib.ac.cn (K.-L.W.); lilin@scib.ac.cn (L.L.); xhzhang@scib.ac.cn (X.-H.Z.); magh@scib.ac.cn (G.-H.M.); 2University of Chinese Academy of Sciences, Beijing 100049, China; 3Independent Researcher, P. O. Box 7, Miki-cho Post Office, Ikenobe 3011-2, Kagawa-ken 761-0799, Japan; jaimetex@yahoo.com; 4Guangdong Provincial Key Laboratory of Applied Botany, South China Botanical Garden, Chinese Academy of Sciences, Guangzhou 510650, China

**Keywords:** *P.* Maudiae, ovule development, ovary injection, genetic transformation, CLSM

## Abstract

In this paper, the development of the *Paphiopedilum* Maudiae embryo sac at different developmental stages after pollination was assessed by confocal laser scanning microscopy. The mature seeds of *P.* Maudiae consisted of an exopleura and a spherical embryo, but without an endosperm, while the inner integument cells were absorbed by the developing embryo. The *P.* Maudiae embryo sac exhibited an *Allium* type of development. The time taken for the embryo to develop to a mature sac was 45-50 days after pollination (DAP) and most mature embryo sacs had completed fertilization and formed zygotes by about 50–54 DAP. *In planta* transformation was achieved by injection of the ovaries by *Agrobacterium*, resulting in 38 protocorms or seedlings after several rounds of hygromycin selection, corresponding to 2, 7, 5, 1, 3, 4, 9, and 7 plantlets from *Agrobacterium*-mediated ovary-injection at 30, 35, 42, 43, 45, 48, 50, and 53 DAP, respectively. Transformation efficiency was highest at 50 DAP (2.54%), followed by 2.48% at 53 DAP and 2.45% at 48 DAP. Four randomly selected hygromycin-resistant plants were GUS-positive after PCR analysis. Semi-quantitative PCR and quantitative real-time PCR analysis revealed the expression of the *hpt* gene in the leaves of eight hygromycin-resistant seedlings following *Agrobacterium*-mediated ovary-injection at 30, 35, 42, 43, 45, 48, 50, and 53 DAP, while *hpt* expression was not detected in the control. The best time to inject *P.* Maudiae ovaries *in planta* with *Agrobacterium* is 48-53 DAP, which corresponds to the period of fertilization. This protocol represents the first genetic transformation protocol for any *Paphiopedilum* species and will allow for expanded molecular breeding programs to introduce useful and interesting genes that can expand its ornamental and horticulturally important characteristics.

## 1. Introduction

*Paphiopedilum* is one of the most exquisite orchid genera with a peculiar flower shape, including pocket-shaped lips, leading these orchids to be commonly referred to as the slipper orchids [1,2]. There are about 107 species in the *Paphiopedilum* genus that are distributed in tropical Asia to the Pacific islands, including 31 species that are found in China, all of which have ornamental value as cut flowers or pot plants [3,4,5]. Due to human-induced destruction of natural habitats and over-harvesting, wild orchid species are under a constant threat of depletion and extinction. The trade of all wild *Paphiopedilum* orchid plants, which are listed in the Convention on International Trade in Endangered Species of Wild Fauna and Flora (CITES) Appendix I, is prohibited [6,7]. Therefore, there is tremendous urgency and need to find solutions for the protection and sustainable utilization of *Paphiopedilum* resources. *P.* Maudiae, which is a cross-fertilized product of *P. callosum* (♀) and *P. lawrenceanum* (♂), has flowers whose color can range from dark red, red, green, to white green, and is a popular ornamental variety in Chinese and international markets due to its high ornamental value and strong adaptability to multiple environments [8]. However, due to limited success in tissue culture protocols from ex vitro-derived explants because of the rarity of materials, difficulties caused by bacterial and fungal decontamination, and the poor development of explants that survive under in vitro conditions [7,9,10], commercial *Paphiopedilum* propagation by growers has, to date, still been entirely through asymbiotic seed germination [2,7,11].

Generative propagation is an important period of growth and development in angiosperms. Ovules form and develop in fruits, and the normal development of ovules is a prerequisite for embryo and seed development. Seed quality is determined by the development of the embryo sac before fertilization, and the development of the embryo and endosperm after fertilization [12]. Since no studies exist on the embryonic development of *P.* Maudiae, this study aimed to research the reproductive biology of this orchid.

There are many methods to assess embryo sac and embryonic development in angiosperms, such as by paraffin and semi-thin sections that are then observed by a light or electron microscope [12,13,14,15,16,17,18]. Confocal laser scanning microscopy (CLSM) is a simple and convenient method [19]. CLSM allows for optical sectioning of live or fixed cells and tissues to obtain information in each layer, like a “microscopic CT image”. In addition, samples can be continuously scanned by CLSM to obtain a three-dimensional structure by computer processing that allows for a clear understanding of the spatial distribution of cells in samples [20]. Even though some reports exist on the use of CLSM to examine embryo sac formation and development in cotton (*Gossypium*) [20], rice (*Oryza sativa*) [21,22], cherry (*Prunus avium*) [23], and some other crops, the technique has not yet been applied to any orchid, including *Paphiopedilum*. In this paper, using CLSM, the reproductive biology of *P.* Maudiae was examined to better understand embryo sac and embryonic development.

Cross breeding in *Paphiopedilum* results in high sterility and a low seed germination rate while the vegetative growth phase of seedlings is long [7,11,24]. Since the success of a breeding program depends on the fine balance between desired and undesirable traits, it is difficult to obtain ideal hybrids. In the last two decades, significant developments in plant transformation technologies aimed at inserting foreign DNA into orchid genomes have been made, and the most popular techniques for genetic transformation are mediated by *Agrobacterium tumefaciens* or microparticle bombardment (biolistics) [25]. In most orchids, protocorm-like bodies (PLBs) were used as target tissue for gene transformation since the origin of these PLBs are single somatic cells, and since PLBs are also easy to root and develop into plantlets [26]. However, they are not suitable for *Paphiopedilum* because it is very difficult to establish a callus or PLB-based regeneration system for *Paphiopedilum*. To date, there are no reports on the genetic transformation of *Paphiopedilum*. Nevertheless, the pollen-tube pathway method [27], which bypasses conventional regeneration, is a relatively simple procedure for genetic transformation [28], and the procedure is very cost effective [16]. The success of *Agrobacterium*-mediated ovary-injection (AMOI) with target genes or plasmids in other orchids (*Dendrobium nobile* and *Doritis pulcherrima*) has been reported by this paper’s corresponding author’s research group who found that transformation efficiency was significantly affected by the period of ovary-injection [29,30].

*Pahiopedilum* orchids usually have a long juvenile phase [31]. *FT* (*Flowering Locus T*) functional genes are key genes that determine flowering time in the photoperiodic pathway and play a very important role in the regulatory processes in flowering plants that underlie the transition from vegetative to reproductive growth [32,33]. Cloning *FT* genes and transforming them into *Pahiopedilum* could elucidate their role during floral bud development and shorten the juvenile phase. In this paper, we established a transgenic protocol for *Pahiopedilum* using the exogenous *CeFT* (*Cymbidium ensifolium Flowering Locus T*) functional gene by AMOI. The ability to obtain transgenic *Paphiopedilum* plants would allow for the creation of new varieties through the introduction of horticulturally useful and ornamentally attractive genes and lay the foundation for genetic breeding research programs for this orchid.

## 2. Results

### 2.1. Embryo Sac Development, Megasporogenesis, and Female Gametophyte Formation

When *P.* Maudiae flowered (Appendix A), just prior to pollination, the placenta was fully developed. At about 6 DAP, the ovaries began to swell and the placenta formed many protuberances, then lengthened and developed into many branches. At about 10 DAP, ovular primordia in the terminus of the branches of the placental protuberances enlarged and differentiated into archesporial cells, which were distinguished by their large nucleus and rich cytoplasmic content. Each was composed of an axial row of cells covered by an epidermal cell layer (Figure 1a). The funicle, which consisted of 4 to 5 cells at the base of the archesporial cell, began to bend and form anatropous ovules until fertilization (Figure 1b). Ovules with a single layer of integument consisted of two layers of cells: Epidermal cells at the bottom of the archesporial cell that then grew to cover the nucellus, leaving the micropyle on top. Nucellus cells developed and formed a single layer of epidermal cells on the outside of the archesporial cells, which differentiated into megasporocytes by elongating and enlarging. Their cytoplasm of megasporocytes was very thick, their nucleus was obvious, and most deviated from the center, very close to the micropyle on top (Figure 1c). As cell volume increased, nucellus cells became gradually compressed. When megasporocytes divided, nucellus cells gradually disintegrated, leaving only the nucellus cells of the chalaza, and thus providing nutrition for the growth of the embryo sac. At about 25 DAP, megasporocytes divided and formed two uniform dyads (Figure 1d). At 25 to 30 DAP, one of the dyads near the micropyle degenerated, while the other near the chalaza continued to divide to form two megaspores of unequal size. The smaller microspore was near the degraded micropyle while the larger microspore, which was near the chalaza, developed into a functional megaspore (Figure 1e). At 35 to 40 DAP, the functional megaspore formed two haploid megaspore nuclei by mitotic division, and since a cell wall did not form between the nuclei, this was referred to as the two-nucleate embryo sac stage (Figure 1f). At 42 to 44 DAP, a four-nucleate embryo sac formed (Figure 1g). At 45 to 50 DAP, eight nuclei formed, and this period was referred to as the eight-nucleate embryo sac, which had an egg cell near the micropyle (Figure 1h), and an antipodal cell near the chalaza (Figure 1i). The two nuclei from the micropyle and chalaza moved to the center of the embryo sac, forming the polar nucleus of the central cell (Figure 1h). The embryo sac matured at about 45–50 DAP while three antipodal cells formed by 54 DAP (Figure 1i).

### 2.2. Fertilization

During the period of fertilization, generative cells divided into two sperm cells, pollen tubes passed a degenerated synergid along the micropyle, entered into the embryo sac and released the two sperm cells which fused with the egg and nucleus, thereby completing double fertilization. The time interval was 50–54 days from pollination to fertilization, and fusion of the pole nuclei formed secondary nuclei before fertilization that formed two endosperm nuclei after fertilization (Figure 2a–c), moving to opposite poles of the embryo sac. One endosperm nucleus was close to the zygote while the other was close to the embryo sac’s chalaza (Figure 2d). In most cases, during embryonic development, the endosperm nuclei no longer divided and gradually degenerated during a later developmental period.

### 2.3. Embryonic Development

The embryo sac of *P.* Maudiae developed to maturity by about 54 DAP when most ovules had become fertilized, becoming zygotes (Figure 3a). As zygote volume increased, polarity strengthened, the cell nucleus was located in the chalaza, which had a nucleolus, and the cytoplasm became very thick. As the embryo developed, the endosperm cell nucleus disintegrated and finally disappeared, leaving mature seeds with no endosperm. A small number of unfertilized ovules did not produce zygotes, and their embryo sac chambers had no embryos. Zygotes underwent uneven horizontal mitosis forming two cells, namely, the basal cell near the micropyle and the apical cell near the chalaza. The volume of the basal cell was larger, it had less cytoplasm and a large vacuole while the volume of the apical cell was small, and the cytoplasm was thicker than that of the basal cell, although the development of basal and apical cells was asynchronous. At 60 DAP, a two-cell pre-embryo stage emerged (Figure 3b). Thereafter, the apical cell formed the embryo after many cell divisions while the basal cell grew into the suspensor in subsequent development. At 70 DAP, the apical cell divided into a linear three-cell pre-embryo (Figure 3c). The basal cell did not divide at this time, and remained vacuolized. At 76 DAP, the cell at the top of the three-cell pre-embryo formed a four-cell T-shaped pre-embryo (Figure 3d). A six-cell embryo formed by periclinal division at 85 DAP, at which time the cytoplasm of the basal cell was thicker, and the cell began to divide and gradually formed the suspensor (Figure 3e). At 90 DAP, the six-cell pre-embryo, following periclinal divisions, formed an eight-cell embryo with a two-cell suspensor with highly vacuolized suspensor cells (Figure 3f). At 100 DAP, the pre-embryo developed into an early globular 16-cell embryo whose suspensor had 2-4 closely arranged cells (Figure 3g). At 105 DAP, the pre-embryo developed into a globular embryo whose suspensor had two cells that showed a high degree of vacuolization (Figure 3h). At 109 DAP, the globular embryo occupied the entire embryo sac chamber, and starch and lipid globules continued to accumulate. The suspensor started to degenerate (Figure 3i). At 114 DAP, the globular embryo remained the same, while the suspensor, which had three cells that were highly vacuolized, gradually degenerated (Figure 3j). At 120 DAP, there was a large number of starch and lipid globules (possible nutrition for seed germination in this stage), but the suspensor had not completely degenerated, and some residue was observed (Figure 3k). At 125 DAP, starch and lipid globules accumulated further (Figure 3l). At 130 DAP, the embryo was globular with a residual suspensor (Figure 3m). At 140 DAP, the testa had wrapped around the globular embryo (Figure 3n). At 150 DAP, the ellipsoid embryo was mature with a large number of starch and lipid globules (Figure 3o). At 160 DAP, the suspensor had degraded completely (Figure 3p). At 170 DAP, the shape of the embryonic body remained unchanged (Figure 3q). At 180 DAP, there were still starch and lipid globules in the embryo (Figure 3r). A similar structure was observed in two other orchids, *Paphiopedilum armeniacum*^18^ and *Cypripedium foemosanum* [34]. In addition, in the process of embryo development, an inner layer of integument cells was absorbed, leaving a layer of cells that developed into the testa (Figure 3b–g). The ovules became overgrown with placenta in three carpels and the ovary had a free cavity at an early developmental stage (from 10 to 90 DAP) (Appendix A) that filled as development progressed (this was observed when capsules were cut for aseptic sowing). The developmental process of the *P.* Maudiae megagametophyte and zygotic embryo is summarized in Table 1.

### 2.4. Morphological Characteristics and Seed Viability Tests

Mature *P.* Maudiae capsules were green (Appendix A) and the seeds were yellow-brown, with an average length of 0.33 mm, fusiform of subacute at both ends, one end with a tip while the other end is blunt. There is a small ellipsoid embryo without an endosperm, and it is only composed of a one-layer testa and an ellipsoidal embryo. The testa is rectangular with a thin cytoderm, brown, transparent, and without any content (Figure 4a). The embryo of *P.* Maudiae seeds did not differentiate completely and consisted only of a mass of undifferentiated embryonic cells, which was the pre-embryo stage. Seeds have no endosperm and do not store nutrients, so they are difficult to germinate in natural conditions without the help of a symbiotic fungus [25]. The embryos of *P.* Maudiae seeds could be stained by triphenyl tetrazolium chloride (TTC) (Figure 4b): If the seeds had an incomplete embryo, seeds were only slightly stained (Figure 4c) or could not be stained in seeds without an embryo (Figure 4d), the embryos of viable seeds were red or light red (Figure 4b,d), while the embryos of aborted seeds did not stain, or stained slightly (Figure 4d,e). However, some seeds with an embryo were not stained (Figure 4d), as was also reported by Fu et al. [11].

### 2.5. In Planta Transformation by Agrobacterium-Mediated Ovary-Injection

The percentage of seeds that stained positive for TTC from the non-injected fruits was 60.61%, which was significantly higher than that of injected fruits at all stages (DAPs), while the percentage of stained seeds in injected fruits at 50 DAP (43.01%, Figure 5d,e) was significantly higher than that at other injected stages, except at 53 DAP (32.98%, Table 2).

### 2.6. TTC and GUS Staining and Preliminary Detection of Transgenic Seeds

The seeds of non-injected fruits by AMOI did not stain GUS-positive. However, some seeds and placentas of injected fruits were stained by GUS (Figure 5d,e), but the level of GUS staining was extremely low (<0.1%, data not shown). Simultaneously, the seeds of injected fruits were used to detect the GUS gene by PCR analysis, which showed a positive band for this gene (Figure 5f), suggesting that this exogenous gene was transferred to *P*. Maudiae seeds.

### 2.7. In Vitro Germination

In the pericarp of mature capsules used in AMOI and the control (no injection), we only sterilized by dipping in 75% (*v*/*v*) ethanol for 1 min, followed by agitation for 15 min in 0.1% (*w*/*v*) mercuric chloride (HgCl_2_) solution, and seeds were then sown onto H26 medium [35]. In this case, seed contamination was 100% for fruits that were injected by *Agrobacterium*, or by other fungi or bacteria, but fruits of the control group were not contaminated. Consequently, we first sterilized the pericarp of mature capsules (treatment and control) using the steps indicated above, then disinfected the seeds after splitting the capsules with a scalpel using NaOCl (0.5% available chlorine) for 40 min. The level of contamination of injected seeds decreased to 20% while no contamination was observed in the control group. Uncontaminated seeds in media were used in follow-up experiments.

Seed germination rate differed at different injection stages in the AMOI treatment and control. Seed germination of the control was highest (36.30%) and significantly higher than seeds at all injection stages, while the highest seed germination of injected fruits was 15.21% at 50 DAP (Table 3).

### 2.8. Selection Pressure and Hygromycin Screening

Uniform sized uninjected protocorms were subcultured on H26 medium containing different concentrations of hygromycin (Hyg). Highest survival (95.33%) was observed on medium without Hyg. As the concentration of Hyg increased, protocorm survival declined, but the rate of decline differed. Highest rates of decline were observed between 5 and 25 mg L^−1^, and between 60 and 75 mg L^−1^, while all protocorms died at 150 mg L^−1^ (Figure 6). A selection pressure gradient was suitable for screening Hyg resistance by successively transferring protocorms to different concentrations of Hyg (25, 40, 40, 25, then 0 mg L^−1^), with each sub-culture period being about 45 d.

Thirty-eight seedlings (with 3–4 leaves and 2–3 roots) were obtained after four rounds of Hyg selection and a single round of Hyg-free culture, resulting in 2, 7, 5, 1, 3, 4, 9, and 7 Hyg-resistant plantlets with a corresponding transformation frequency of 0.45%, 1.89%, 2.16%, 1.07%, 0.90%, 2.45%, 2.54%, and 2.48% from protocorms which germinated from AMOI seeds at 30, 35, 42, 43, 45, 48, 50, and 53 DAP, respectively. No Hyg-resistance plantlets formed from seeds of uninjected capsules or from injected capsules older than 55 DAP. The highest transformation frequency was obtained at 48, 50, and 53 DAP, which were significantly higher than other AMOI times. The highest transformation frequency of all sown seeds was obtained at 50 DAP, which was significantly higher than other AMOI times, except at 35 and 53 DAP (Table 4). Hyg-resistant plantlets transferred to Hyg-free H26 media could form complete plants.

### 2.9. GUS Staining and Molecular Detection of Hyg-Resistant Plants

The leaves of all putatively Hyg-resistant plants stained blue after GUS staining (Figure 5c). Veins were stained most intensely, indicating that the GUS stain may be easier to permeate into vein tissue. Four randomly selected Hyg-resistant plants were subjected to PCR analysis, and positive bands for the *GUS* and *FT* genes were observed (Figure 5g). PCR analysis and semi-quantitative PCR were used to determine the presence of the *hpt* gene and its expression in leaves of Hyg-resistant seedlings from AMOI at 30, 35, 42, 43, 45, 48, 50, and 53 DAP (Figure 5h,i). *hpt* expression was detected by quantitative real-time PCR analysis in these tested seedlings as previously described, but at different levels, indicating that exogenous genes were integrated into the plant genome (Figure 5j). Transformation efficiency at 50 DAP was highest (2.54%), followed by 2.48% at 53 DAP and 2.45% at 48 DAP (Table 4). The best time to inject *P.* Maudiae fruits using the *in planta* AMOI method for genetic transformation was 48–53 DAP.

## 3. Discussion

### 3.1. Ovules and Embryo Development

Seed germination rates and genetic transformation efficiency by ovary-injection in orchids are significantly affected by the embryo developmental stage [2,18,30]. The period of fertilization, and the time it takes for pollen to fall on the stigma and induce the fusion of its sperm with the eggs and polar nucleus differs in most plant species, and the process tends to be complete in 10-48 h [36,37]. However, most orchids need more time, usually 40 to 90 d [12,14]. In this paper, we found that the time required from pollination to fertilization in *P.* Maudiae is about 50 to 54 d. In *P.* Maudiae flowers, the ovaries are small, but by about 21 DAP, the capsules expand rapidly, which may be due to the growth of pollen tubes in the ovaries. Pollen inclusions may have some physiological regulatory substances, including auxin and ethylene, which can promote ovule development, seed coat formation and the maturation of female gametophytes [12,38]. The exact reason is unclear, however, and the main active substances and the mechanism of action need to be further researched.

During the development of the female *P.* Maudiae gametophytes, only the megasporocytes were surrounded by a layer of nucellus cells, thus *P.* Maudiae is a thin nucellus-type plant. Based on a reference standard of angiosperm embryo sac development types, the *P.* Maudiae embryo sac development is classified as *A**llium* type [39]. The process of zygote division and differentiation into embryos proceeded according to standard patterns, and the difference in embryonic development between angiosperms are the first few cell divisions of the zygote [40]. Embryonic development in *P.* Maudiae first shows a transverse division in which the topical cell divides transversally and the basal cell is involved in the formation of the embryonic body and division to form suspensor cells. Then, germ cells develop into zygotes, which continue to develop and do not undergo dormancy, unlike in *Paphiopedilum godefroy* [41] in which zygotes begin to divide after 40–45 d of dormancy. Based on the reference standard of angiosperm embryonic development types, *P. godefroy* embryonic development is a *S**olanum* type [39,40].

Embryonic development of mature *P.* Maudiae seeds generated some seeds without embryos, which may be due to two reasons: (1) Development of the female gametophyte is not normal during embryo sac development, and a mature embryo sac does not form; (2) only a part of the embryo sac can fertilize normally and form a zygote. However, development of the male gametophyte and the processes of pollination and fertilization require further research.

Development of the embryo sac is the process in which the female generative organ of a plant forms, and is an important aspect of reproductive biology. Normal paraffin sections and semi-thin sections are currently the most common methods to observe embryo sacs and embryo development in orchids [13,14,15,16,17,18,41]. The procedural preparation for these two methods is long and tedious, and it is very difficult to isolate several nuclei in embryo sac development within the same time section. CLSM is a combination of common optical microscopy and computer-aided laser technology whose software allows for continuous optical sectioning, allowing for the observation of dynamic changes to the cytoskeleton and other structures at the subcellular level [19]. CLSM significantly shortens the operation time, making it more convenient, and also allowing different levels of organizational structure to be accurately observed. A series of images allows for 3D reconstruction to observe the internal structure and space relationships from different angles [20].

In this paper, treated *P.* Maudiae embryo sacs and embryo development were observed by CLSM, and clear pictures of both were obtained. This allowed us to determine the developmental stages of embryo sacs and embryos at different periods after pollination, laying down a foundation for further research on developmental studies in this orchid.

### 3.2. Agrobacterium-Mediated Ovary-Injection

Currently, the most common ways to transform orchids are by *Agrobacterium* [25,42,43,44,45,46] and particle bombardment [47,48,49,50,51,52], as has been reviewed [53]. The pollen tube pathway was successfully used for transgenic breeding in corn, wheat, melons, cucumbers and other crops such as cotton [28,54,55,56,57]. However, there are few reports for orchids because most orchids contain an abundance of seeds in each capsule that cannot germinate under natural conditions. Orchid seeds thus need an artificial regeneration system to be established, either symbiotically [58], or asymbiotically [59], as has been shown for another orchid genus, *Dendrobium*. Moreover, the plasmid or DNA may degrade easily which could be problematic because most orchids generally require 40-90 days from pollination to fertilization [12]. Nevertheless, the fruits of most orchids are large, and the ovary has large cavities that can hold in excess of 10,000 seeds, allowing more injection solution to be loaded for transformation.

Thus, the injection of *Agrobacterium* carrying a plasmid with DNA of interest into the ovary may provide an effective way to introduce novel genes into orchids. Two preliminary studies recorded success in other orchids by some of the authors of this paper, namely in *Dendrobium nobile* [29] and *Doritis pulcherrima* [30].

An appropriate selective marker is important for an effective resistance screening test. In this experiment, the Ti plasmid contained the Hyg resistance gene. Orchid protocorms and seedlings are not particularly sensitive to kanamycin, but effective screening can be achieved when using Hyg, as in our study, and which was also shown to be suitable for other orchids, including *Dendrobium nobile* [29,59] and *Doritis pulcherima* [30]. The appropriate selection pressure (i.e., Hyg concentration) is also important in resistance screening, and if the Hyg concentration is too low, untransformed protocorms will continue to grow and divide, and putative transformants may be lost. If, on the other hand, the Hyg concentration is too high, protocorm growth will be suppressed or they may even die, even killing off transformed cells. The selection gradient employed for screening transgenic protocorms in this experiment was suitable. A total of 38 Hyg-resistant plants with 3 to 4 leaves and 2 to 3 roots were obtained after four cycles of Hyg resistance screening and a single round of Hyg-free culture, the leaves of all resistant plants stained GUS-positive, and four randomly selected Hyg-resistant plants tested positive for the *GUS* and *FT* genes by PCR analysis. These results show that these two exogenous genes could be integrated into the *P*. Maudiae genome. Semi-quantitative PCR and quantitative real-time PCR analysis of total RNA isolated from the leaves of Hyg-resistant seedlings demonstrates the same result, i.e., confirms the integration of these genes. However, the stable inheritance of these exogenous genes in subsequent generations, and whether the *FT* gene will shorten the juvenile stage when overexpressed still need to be researched. In general, *P*. Maudiae plants need about three years to flower from in vitro seedlings, so there is hope that plants transformed with the *FT* gene may flower precociously. This is because the biological function of *FT* is a putative phosphatidylethanolamine-binding protein gene, as was shown in *Cymbidium*, that may regulate the vegetative to reproductive transition in flowers and act similarly to *AtFT* by regulating the transition from vegetative state to flowering by activating *AtAP1* [60,61,62].

The period of exogenous DNA transformation by the pollen tube pathway using transgenic technology is limited to the time between fusion of the sperm and egg and division of the zygote [28,63,64,65,66]. The exogenous gene enters the embryo sac via the pollen, placental surface, micropyle and nucellus channel, and can directly transform the generative cells without a cell wall in the fusion period [67,68,69]. The distance traveled by exogenous DNA before entering the embryo is an important factor that influences transformation frequency [28,29,30]. In our experiments, AMOI provided a shorter entrance pathway for exogenous DNA and thus a shorter entrance period, a greater quantity of DNA entering the embryo sacs, and less degradation of exogenous DNA. A shorter DNA entrance pathway would therefore greatly improve the capacity of exogenous DNA to enter the embryo sacs and enhance the probability of its integration into the host genome. In this study, 38 Hyg-resistant plantlets were obtained from capsules that had been injected between 30 and 53 DAP, with the highest transformation frequency (2.45% to 2.48%) from 48 to 53 DAP. No Hyg-resistant plantlets were obtained when capsules older than 55 DAP were injected. When considering that fertilization in *P.* Maudiae takes place between 50 and 54 DAP, the best time to employ AMOI would then be when double fertilization of the eggs takes place. However, the lower transformation frequency of capsules injected at 43 and 45 DAP than that of an earlier injection period is difficult to explain because less degradation of exogenous DNA occurred in a shorter time frame from injection of exogenous DNA to fertilization. A possible reason for the lower transformation frequency of all sown seeds form ovules injected 43-48 DAP is that the embryo is easily damaged by AMOI when the embryo sac is in the four- to eight-nucleate stage of development. However, we could not detect *CeFT* gene mRNA from AMOI at 30, 35, 42, 43, 45, 48, 50, and 53 DAP, possibly due to the loss or silencing of heterologous *CeFT* gene during seed germination and growth, or because the Ubi promoter was unable to activate the expression of *CeFT* in *P.* Maudiae.

In conclusion, the *P.* Maudiae embryo sac exhibited an *Allium* type of development. AMOI is an attractive and effective method for the genetic transformation of *Paphiopedilum*. The fertilization time of *P.* Maudiae was 50 to 54 DAP, and the best injection time was coincidentally 48 to 53 DAP, which corresponds to the period of fertilization. Thirty-eight protocorms or seedlings were obtained after several rounds of Hyg selection and four randomly selected Hyg-resistant plants were GUS-positive. Semi-quantitative PCR and quantitative real-time PCR analyses revealed the expression of the *hpt* gene in the leaves of eight Hyg-resistant seedlings. This protocol represents the first effective genetic transformation protocol for any *Paphiopedilum* species and will allow for expanded molecular breeding programs to introduce useful and interesting genes that can expand its ornamental and horticulturally important characteristics.

## 4. Materials and Methods

### 4.1. Plant Material and Agrobacterium Strain

*Paphiopedilum* Maudiae plants were maintained in a glass greenhouse with a water curtain and blowers for cooling and ventilation in the South China Botanical Garden, Guangzhou, China. The average temperature and humidity range were 15–35 °C and 65–95%, respectively. The flowers from three-year-old adult plants were labeled and artificially self-pollinated by transferring pollen onto the stigma of the same flower as they became fully opened in March to April, in 2014. The seed setting percentage exceeded 90%.

The *A. tumefaciens* EHA105 strain carried the *Flowering Locus T gene* (*CsFT*) from *Cymbidium sinense*, the β-glucose acid gene (*GusA*), the Hyg resistance gene (*hpt*) with a 35S CaMV promoter and a NOS terminator [60].

### 4.2. Histological and Histochemical Studies

Fruit capsules (Appendix A) of different developmental stages from 5 to 180 DAP were collected at 5- or 10-d intervals. Capsules were removed and both ends were sliced horizontally, then fixed in a solution of FAA (50% ethanol:acetic acid:formalin; 18:1:1, *v*/*v*) at 4 °C overnight. Ovules, which were excised from fixed capsules, were stained according to a previous protocol [22], as follows: (1) A rinse with double distilled water six times, each time for 20 min; (2) rehydration in an ethanol gradient with 50%, 70%, 90% ethanol, each time for 20 min; (3) staining with solution (95% ethanol, 5% eosin, and 5% golden orange; 96:3:1, *v*/*v*) for 21 h; (4) rinse in double distilled water six times, each time for 20 min; (5) dehydration in an ethanol gradient with 50%, 70%, 90% ethanol, each time for 20 min, then dehydrated twice with 100% ethanol, each time for 30 min; (6) treatment in a mixed solution (100% ethanol, methyl salicylate; 1:1, *v*/*v*) for 3 h, then added to methyl salicylate three times (first time = 3 h; second and third times = 15 h each). Transparent ovules were continually preserved in methyl salicylate; (7) whole transparent ovules were mounted on microslides, covered with coverslips (microslides and coverslips: Leica Biosystems Inc., Buffalo Grove, IL, USA), and observed and photographed at 543 nm laser excitation using a laser scanning confocal microscope (Lsm 510 Meta, Zeiss, Bonn, Germany) for CLSM studies.

### 4.3. Injection of Ovaries with Agrobacterium

In the glass greenhouse, 20 µL of *Agrobacterium* suspension was injected laterally into the middle of capsules (ovaries) using a 25 µL microinjector (Shanghai Eastsen Analytical Instrument Co. Ltd., Shanghai, China) on 30, 35, 40, 42, 43, 44, 45, 46, 47, 48, 50, 53, 55, 57, 60, 65, and 70 DAP, i.e., a total of 17 time treatments, with nine different capsules per time treatment. The control consisted of capsules that were not injected with *Agrobacterium* after artificial pollination. Seed from 150 DAP fruits were sown aseptically.

### 4.4. Aseptic Sowing

Capsules that had (or had not) been injected with *Agrobacterium* were aseptically sown, as follows. At first, capsules were rinsed under running tap water to remove remains of the epidermis and disinfected by dipping into 75% (*v*/*v*) ethanol for 1 min, then immersed for 15 min in 0.1% (*w*/*v*) mercuric chloride (HgCl_2_) solution. Capsules were then rinsed four times with sterilized (by autoclave) deionized water and their surfaces were dabbed dry on sterilized filter paper to remove surface moisture. Capsules were split longitudinally along the suture with a sterilized scalpel, seeds were gently placed into sterilized 100-mL conical flasks for sterilizing with NaOCl containing 0.5% available chlorine for 40 min. Seeds were then rinsed three times with sterile deionized water (filtered through sterilized filter paper; Hangzhou WoHua Filter Paper Co. Ltd., Hangzhou, China) and placed into sterile deionized water to create a suspension liquid with approximately 100 seeds/mL for aseptic sowing.

Three ml of a suspension with about 300 seeds from all injection stages were dispersed uniformly on Hyponex N026 medium [2], which contained 1.5 g L^−1^ Hyponex I (Taihe Horticultural Co. Ltd., Taiwan, China), 15 g L^−1^ sucrose, 2 g L^−1^ peptone, 5.5 g L^−1^ agar (Huankai Microbial Sci. & Tech, Co., Ltd., Guangzhou, China), 1.5 g L^−1^ activated charcoal (AC), 10 mg L^−1^ vitamin B_1_, 1 mg L^−1^ vitamin B_6_, 1 mg L^−1^ niacin, 2 mg L^−1^ glycine, 1 mg L^−1^ NAA, 1 mg L^−1^
*myo*-inositol, and 50 mL L^−1^ coconut milk (CM). All chemicals and reagents were purchased from Sigma-Aldrich (St. Louis, MI, USA), unless specified otherwise. The CM used in these experiments was obtained from 6- to 7-month-old green coconuts from Hainan Province, China, after filtering through a single sheet of filter paper [35]. Medium was added to 500-mL conical flasks (Xuzhou Hualian Glass Products Co. Ltd., Xuzhou, China) closed with perforated rubber stoppers and plugged with cotton. Each flask contained about 100 mL of medium. Medium pH was adjusted to 5.6 with 1 mol L^−1^ KOH or HCl before autoclaving at 121 °C for 18 min at 1.06 kg cm^−2^. Cultures were incubated at 26 ± 1 °C and a 16-h photoperiod under 40 W cool white fluorescent lamps (Shanghai Xianyi Lighting and Electrical Appliance Co. Ltd., Shanghai, China) delivering a photosynthetic photon flux density of ca. 45 μmol m^−2^ s^−1^. The germination rate of seeds was calculated using a previously reported method [2].

### 4.5. Seed Viability Test

Mature *P.* Maudiae seeds at all tested DAP were added into labelled 1.5-mL Eppendorf tubes. Approximately 100 seeds were submerged in 1.0 mL of 0.5% triphenyl tetrazolium chloride (TTC) solution, incubated at 37 °C, stained for 24 h, then rinsed with deionized water. TTC solution was made by first dissolving 0.1 g of TTC in 200 mL phosphate buffer (pH 7.17), then placing the TTC solution into an amber laboratory bottle at 4 °C. Stained seeds were viewed with a stereomicroscope (Leica DFC 450, Wetzlar, Germany) to count the number of seeds with an embryo and the number of stained seeds. Measurements were repeated three times for each treatment (i.e., DAP) at all injection stages.

### 4.6. GUS Assay for Seeds and Transgenic Seedlings

Mature *P.* Maudiae seeds (150 DAP) at each injection stage were placed into separate Eppendorf tubes. Seeds were imbibed (no vacuum infiltration) in GUS solution (100 mM L^−1^ sodium phosphate buffer (Na_3_PO_4_), 0.1% Triton X–100, 10 mM l^−1^ EDTA, 0.5 mM l^−1^ potassium ferricyanide, 0.5 mM L^−1^ potassium ferrocyanide, 1 mg ml^−1^ X-Gluc), placed in an incubator and stained for 12 h at 37 °C, destained in 75% ethanol for 1-2 d, then placed on a microslide and viewed by a Leica DFC 450 stereomicroscope. The leaves of putatively transgenic seedlings with 3–4 leaves resistant to Hyg were detected by GUS staining using the same method employed for the GUS staining of seeds.

### 4.7. Selection Pressure and Resistance Screening of Protocorms

Well-developed protocorms of the control group with uniform growth and size (approx. 0.4 cm in diameter) were inoculated onto Hyponex N026 medium containing different concentrations of Hyg (0, 5, 10, 15, 20, 25, 30, 40, 50, 60, 70, 75, 100, 150, and 200 mg L^−1^). Each culture dish (90 mm × 16 mm) was inoculated with 50 protocorms, with five dishes for each treatment. Trials indicated that 25–40 mg L^−1^ was the most effective concentration range for screening the resistance of protocorms to Hyg. Protocorms that survived were cultured successively in media with different concentrations of Hyg (25, 40, 40, 25, and 0 mg L^−1^), each culture cycle being 45 d. The gradually decrease in the concentration of Hyg in the last two rounds was to encourage the growth of transgenic material to develop robust plants.

### 4.8. PCR Detection of Seeds and Resistant Seedlings

*CeFT*, *GUS* and *hpt* genes were detected in seeds at 150 DAP from ovary-injection of *Agrobacterium* at 48 DAP, and the *hpt* gene was detected in Hyg-resistant seedlings from AMOI at 30, 35, 42, 43, 45, 48, 50, and 53 DAP. The DNA of Hyg-resistant seedlings was extracted by the cetyl trimethyl ammonium bromide (CTAB) method [58]. The PCU1301 plasmid, carrying exogenous genes, served as the positive control and untransformed seedlings served as the negative control. A map of the T-DNA region is indicated in Appendix A. The following primers were used for PCR of the *hpt* and *GUS* genes [58], and the *FT* gene [60]: HPTL (forward) 5′-GATGTTGGCGACCTCGTATT-3′; HPTR (reverse) 5′-GTGCTTGACATTGGGGAGT-3′; GUSL (forward) 5′-GTGAATCCGCACCTCTGG-3′; GUSR (reverse) 5′-ATCGCCGCTTTGGACATA-3′; FTL (forward) 5′-CGCGGATCCATGAATAGAGAGAGAGACT-3′; FTR (reverse) 5′-CGCGGATCCTCAATCCTGCATCCTTCTTCCG-3′. PCR was run over 35 cycles of 95 °C for 30 s, 58 °C for 30 s, and 72 °C for 1 min. PCR products were detected by 0.8% agarose gel (Biowest Agarose G-10, Baygene Biotech. Co., Ltd., Shanghai, China) electrophoresis, stained with 0.1 µg ml^−1^ ethidium bromide (Leagene Biotech. Co., Ltd., Beijing, China), observed and images were collected by a UVP GelDoc-It 310 imaging system (Shanghai Yuansheng Instrument Equipment Co., Ltd., Shanghai, China).

### 4.9. Semi-Quantitative PCR and Quantitative Real-Time PCR Analysis

The *hpt* gene was detected in Hyg-resistant seedlings from AMOI at 30, 35, 42, 43, 45, 48, 50, and 53 DAP. Total RNA from these leaves was extracted using a Column Plant RNAOUT 2.0 kit (TIANDZ Biotech. Co., Ltd., Beijing, China), and 1 μg of each RNA sample were reverse transcribed using the GoScript™ Reverse Transcription System (Promega Biotech Co., Ltd., Beijing, China) according to the manufacturers’ instructions. The cDNA products were diluted to 200 ng µL^−1^ for further studies.

The success of reverse transcription was verified by PCR using the *RPS3a* reference gene [70]. The gene-specific primer pairs were designed by the online PrimerQuest tool [71] (RPS3aL (forward) 5′-TGATGTGAAGACCACGGATAAC-3′, RPS3aR (reverse) 5′-CAGGTTCGCTTGACTTGATTTG-3′). Semi-quantitative PCR was performed using the Golden Star T6 Super PCR Mix (TsingKe Biotech Co., Ltd., Beijing, China) with first-strand cDNA as the template in a 25 μL standard PCR reaction. The PCR reaction was performed by denaturation at 95 °C for 2 min, followed by 30 cycles of 10 s denaturation at 98 °C, 10 s annealing at 65 °C, 10 s extension at 72 °C, and a final extension at 72 °C for 1 min. The amplified fragments were separated on a 1% agarose gel containing 0.1 µg mL^−1^ ethidium bromide in TAE buffer.

The cDNA protocol described above was also used for quantitative real-time PCR analysis, which was performed using the iTaq Universal SYBR Green Supermix (Bio-Rad Laboratories, Inc., Beijing, China) in an *Applied Biosystems*^®^ ABI 7500 Real-time system (Thermo Fisher Scientific Inc., Waltham, MA, USA). Amplification conditions were 95 °C for 30 s, followed by 40 cycles of amplification (95 °C for 15 s, 60 °C for 35 s).

### 4.10. Data Analysis

The experiments of TTC staining and seed germination were conducted in a completely randomized design. All data were analyzed with SPSS 17.0 for Windows (Microsoft Corp., Washington, USA) and expressed as means ± standard error (SE) using one-way analysis of variance (ANOVA) followed by Duncan’s multiple range test at *p* ≤ 0.05.

## Figures and Tables

**Figure 1 ijms-22-00084-f001:**
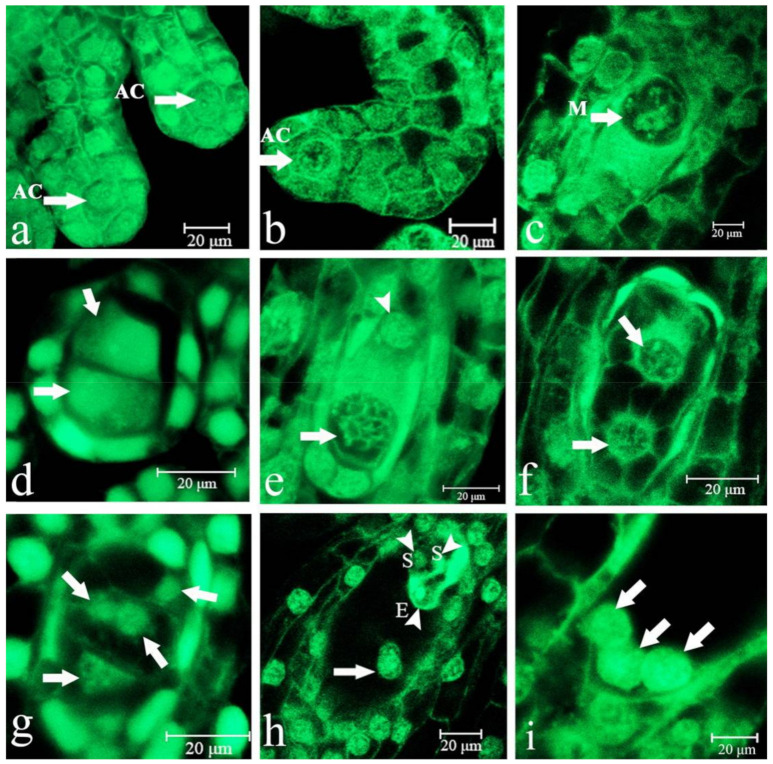
Development of the *Paphiopedilum* Maudiae embryo sac at several stages before pollination. (**a**): Archesporial cell (AC) at 10 days after pollination (DAP) (arrow); (**b**): AC at 20 DAP (arrow) in which the ovule primordium began to curve back to eventually face the placenta; (**c**): Megasporocyte (M) at 25 DAP (arrow); (**d**): First meiotic cell division with the formation of two dyads (arrow) at 25 DAP; the chalazal dyad enlarged prior to the second meiotic cell division; (**e**): Second meiotic cell division formed two megaspores of unequal size at 30 DAP; the smaller megaspore gradually degenerated (arrow); (**f**): First mitotic nuclear division with the formation of a two-nucleate embryo sac (arrow) at 35 DAP; (**g**): Second mitotic nuclear division with the formation of a four-nucleate embryo sac (arrow) at 42 DAP; (**h**): Polar nucleus (arrow), egg cell (E), two synergids (S) at 45 DAP; (**i**): Three antipodal cells (arrow) at 54 DAP.

**Figure 2 ijms-22-00084-f002:**
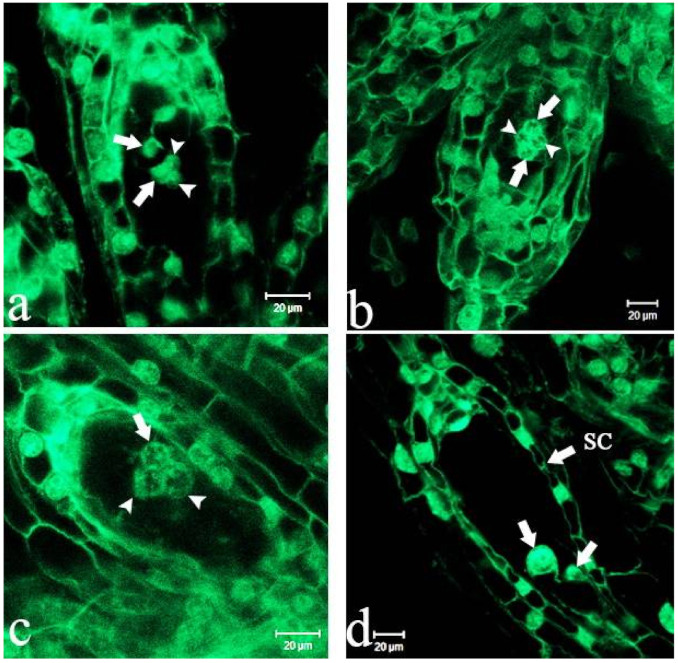
Polar nucleus of *Paphiopedilum* Maudiae embryo sac development after pollination. (**a**–**c**): Fertilization of secondary nucleus; (**d**): Chalazal endosperm nucleus (arrow). SC = seed coat.

**Figure 3 ijms-22-00084-f003:**
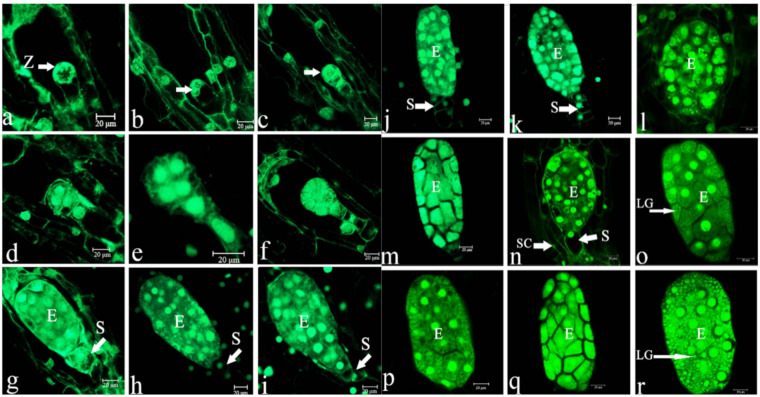
Development of the *Paphiopedilum* Maudiae embryo at different stages after pollination. (**a**): Zygote (Z) just fertilized at 54 days after pollination (DAP) (arrow); (**b**): Two-cell pre-embryo at 60 DAP (arrow); (**c**): Three-cell pre-embryo at 70 DAP (arrow); (**d**): T-shaped pre-embryo with four cells at 76 DAP; (**e**): Six-cell embryo at 85 DAP; (**f**): Eight-cell embryo at 90 DAP; (**g**): Sixteen-cell embryo (E) at 100 DAP, suspensor (S); (**h**): Spherical embryo at 105 DAP; (**i**): Spherical embryo at 109 DAP; (**j**): Suspensor (S) with three highly vacuolated cells at 114 DAP (arrow); (**k**): Suspensor (S) with three highly vacuolated cells had not completely degenerated at 120 DAP (arrow); (**l**): Abundant starch and lipid globule accumulation at 125 DAP; (**m**): The suspensor degenerated, but remained residual at 130 DAP; (**n**): Spheric embryo was very close to the testa at 140 DAP (arrow); SC = seed coat; (**o**): Mature spherical embryo at 150 DAP; (**p**): The suspensor disappeared at 160 DAP; (**q**): Shape of the spherical embryo remained unchanged at 170 DAP; (**r**): Mature spherical embryo with large starch and lipid globules (LG) at 180 DAP.

**Figure 4 ijms-22-00084-f004:**
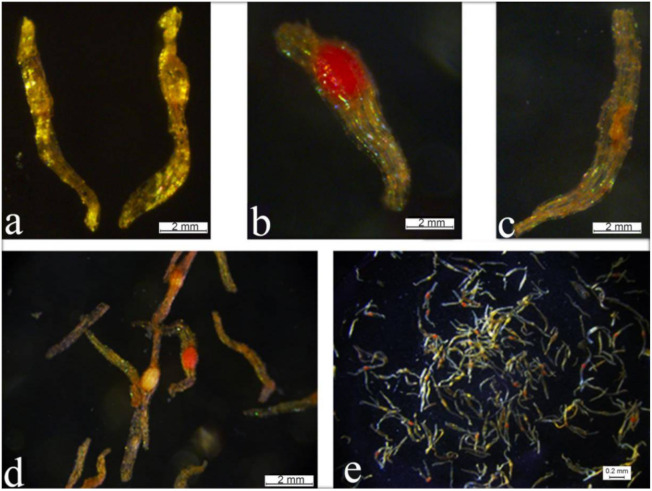
Viability test of *Paphiopedilum* Maudiae seeds. (**a**): Seeds of *P.* Maudiae; (**b**): Triphenyl tetrazolium chloride (TTC) staining of viable seed; (**c**): Slight TTC staining of seeds with an aborted embryo; (**d**): Seeds with an unstained embryo; (**e**): TTC staining of seeds from a capsule.

**Figure 5 ijms-22-00084-f005:**
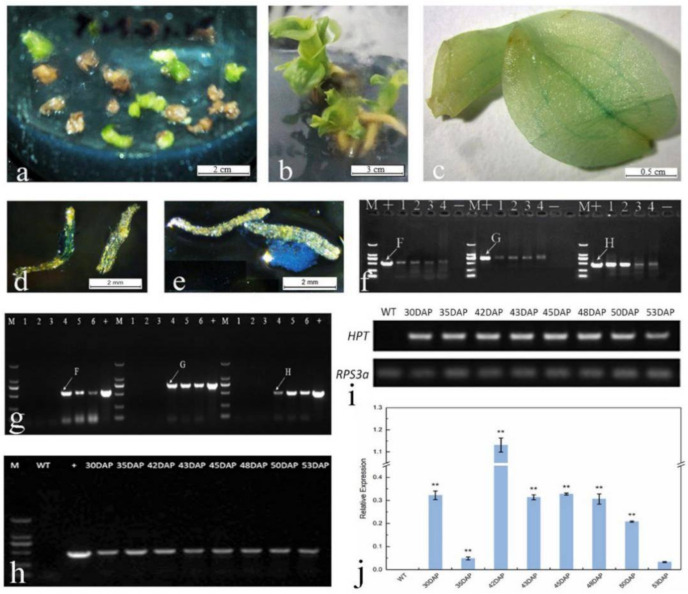
*In pl**anta* transformation of *Paphiopedilum* Maudiae by *Agrobacterium*-mediated ovary-injection (AMOI). (**a**): Protocorms selected on H26 medium [35] with 40 mg L^−1^ hygromycin (Hyg) in second screening; (**b**): Hyg-resistant plants after Hyg selection on H26 medium without Hyg; (**c**): Leaves of Hyg-resistant plants were stained blue after GUS staining; (**d**): GUS expression in the placenta; (**e**): GUS expression in seeds; (**f**): Detection of *CeFT*, *GUS* and *hpt* genes; lanes 1-4, AMOI at 35, 48, 50 and 53 DAP; + lane, positive control;—lane, negative control; M, DL2000 DNA marker ladder (TaKaRa, Dalian, China); (**g**): Detection of *CeFT*, *GUS* and *hpt* genes in 150 DAP seeds after AMOI at 48 DAP; lanes 1, 2 and 3: seeds from no AMOI fruits; lanes 4, 5 and 6: seeds from AMOI fruits; + lane, positive control; M, DL2000 DNA marker ladder; (**h**): PCR analysis of *hpt* in Hyg-resistant seedlings from AMOI at 30, 35, 42, 43, 45, 48, 50, and 53 DAP. M: DL2000 DNA marker ladder, WT: wild-type plant, +: positive plasmid control. (**i**): Semi-quantitative PCR analysis of *hpt* in Hyg-resistant seedlings from AMOI at 30, 35, 42, 43, 45, 48, 50, and 53 DAP. (**j**): *hpt* expression in WT and Hyg-resistant seedlings as revealed by quantitative real-time PCR. Data were calculated using the 2^−ΔΔCt^ formula. Error bars represent standard error of three replicates. Double asterisks (**) indicate a significant difference relative to WT according to Duncan’s multiple range test (*p* < 0.01).

**Figure 6 ijms-22-00084-f006:**
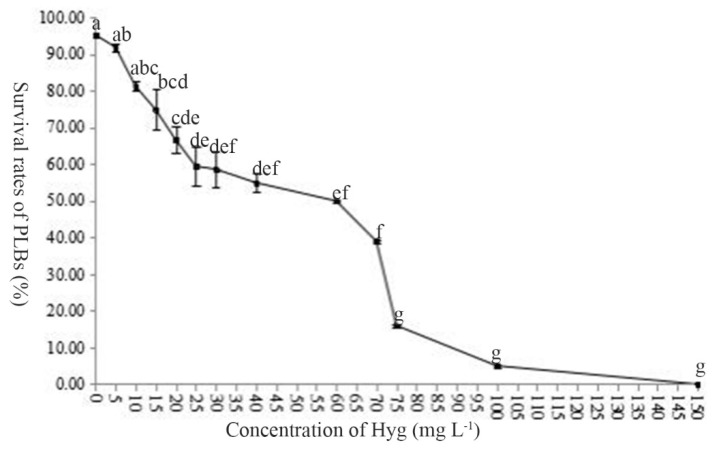
Effect of different Hyg concentrations on *Paphiopedilum* Maudiae protocorms. For each treatment, 50 protocorms were inoculated per culture dish (90 mm × 16 mm). All experiments consisted of five independent replicates. Values followed by different letters indicate significant differences according to Duncan’s multiple range test (*p* < 0.05).

**Table 1 ijms-22-00084-t001:** Major microscopic structural events taking place in the developing megagametophyte and embryo of *Paphiopedilum* Maudiae.

Days after Pollination (DAP)	Developmental Stage
6, 10, 20	Formation of an archesporial cell
20, 25	Formation of megasporocyte
25, 30	Formation of functional megaspore
35, 38, 40	Two-nucleate embryo sac
42, 44	Four-nucleate embryo sac
45, 50	Eight-nucleate embryo sac
50, 54	Mature embryo sac, fertilization
54, 60	First division of the zygote
70, 76, 80, 85, 90	Pre-embryo with 2-8 cells, suspensor with 1–2 cells and two-nucleate endosperm
95, 100, 105, 109, 114	Pre-embryo with 10-40 cells, suspensor with 2–4 cells and nucleate endosperm
120, 125, 130, 140	Embryo with more than 100 cells, suspensor and nucleate endosperm degenerated, massive starch and lipid globule found in the embryonal cells
150	Seed mature
160	Seed fully mature
180	Capsules begin to open

**Table 2 ijms-22-00084-t002:** TTC staining rates of seeds derived from *A**grobacterium*-mediated ovary-injection at different days after pollination (DAP).

Injection Time (DAP)	TTC Staining (%) *
Control (no injection)	60.61 ± 0.05 a
35	12.11 ± 4.41 ef
40	14.63 ± 5.33 def
44	18.63 ± 2.61 cde
46	24.85 ± 3.70 cde
50	43.01 ± 3.60 b
53	32.98 ± 2.47 bc
55	22.31 ± 0.84 cdef
57	19.96 ± 6.79 cd
60	14.04 ± 1.24 def
65	11.91 ± 7.51 def
70	10.43 ± 4.06 f

* For each treatment, approximately 300 seeds were stained by TTC. All experiments consisted of three independent replicates. Values followed by different lower-case letters within the column are significantly different at *p* < 0.05 according to Duncan’s multiple range test.

**Table 3 ijms-22-00084-t003:** Germination rates of seeds, 150 days after pollination (DAP), derived from *Agrobacterium*-mediated ovary-injection at different injection times on H26 medium [35] at 120 days after culture.

Injection Time (DAP)	Seed Germination (%) *
Control (no injection)	36.30 ± 2.13 a
35	7.26 ± 2.50 bc
40	7.53 ± 1.65 bc
44	8.78 ± 1.15 bc
46	10.32 ± 1.26 bc
50	15.21 ± 2.91 bc
53	9.31 ± 0.43 bc
55	5.46 ± 1.23 c
57	4.71 ± 0.10 c
60	4.61 ± 0.91 c
65	3.66 ± 0.93 c
70	2.84 ± 1.08 c

* For each treatment, approximately 300 seeds were cultured in a 500-mL culture flask containing 90 mL of medium. All experiments consisted of three independent replicates with 10 culture flasks per replicate. Values followed by different lower-case letters within the column are significantly different at *p* < 0.05 according to Duncan’s multiple range test.

**Table 4 ijms-22-00084-t004:** Effect on transformation rates of *Paphiopedilum* Maudiae derived from *Agrobacterium*-mediated ovary-injection at different days after pollination (DAP).

Injection Time (DAP)	Number of Hygromycin-Resistant Seedlings	Number of Germinated Seeds	Percentage of Resistant Seedlings (Transformation Frequency (%))	Percentage of Hygromycin-Resistant Seedlings from all Seeds Sown * (Transformation Frequency (%))
No injection	0	530	0 g	0 e
30	2	443	0.45 f	0.13 cd
35	7	371	1.89 c	0.47 ab
42	5	231	2.16 b	0.33 bc
43	1	235	1.07 d	0.07 de
45	3	334	0.90 e	0.20 cd
48	4	163	2.45 a	0.27 c
50	9	354	2.54 a	0.60 a
53	7	282	2.48 a	0.47 ab
55	0	242	0 g	0 e
57	0	230	0 g	0 e

* For each injection time, approximately 1500 seeds were sown. Values followed by different lower-case letters within the column are significantly different at *p* < 0.05 according to Duncan’s multiple range test.

## Data Availability

Data may be found within the article or Appendix A. Raw data available upon reasonable request.

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
