# Peer review of "Ovule Development and in Planta Transformation of Paphiopedilum Maudiae by Agrobacterium-Mediated Ovary-Injection"

_ijms, 2020, doi:10.3390/ijms22010084_

Round 1
Reviewer 1 Report
Dear Authors,
The MS is very well writen. I have minor questions.
Perhaps in the abstract, comment about the importance of techniques for breading in orchid industry/producers and the advance obtained in the present paper.
Line 60. Check the reference style.
Figure 7 reference in the main text in result section, is missing. The authors cite the figure 7 in M & M. Main a comment in Results section, where the figure is presented.
Line 500. Check the Reference style.
Line 553. To add the website to the References section, as a new reference.
Perhaps, conclusions could be more ellaborate. They are too short from my point of view.
Author Response
Response to Reviewer 1
Comments and Suggestions for Authors
Dear Authors,
The MS is very well writen. I have minor questions.
Thank you, we appreciate this opinion.
Perhaps in the abstract, comment about the importance of techniques for breading in orchid industry/producers and the advance obtained in the present paper.
A sentence was added to the Abstract: “This protocol represents the first genetic transformation protocol for any Paphiopedilum species and will allow for expanded molecular breeding programs to introduce useful and interesting genes that can expand its ornamental and horticulturally important characteristics.”
Line 60. Check the reference style.
The correct reference style has been used.
Figure 7 reference in the main text in result section, is missing. The authors cite the figure 7 in M & M. Main a comment in Results section, where the figure is presented.
Figure 7 had been changed to Supplemental information 1, which was cited in the manuscript.
Line 500. Check the Reference style.
The correct reference style has been used.
Line 553. To add the website to the References section, as a new reference.
The website was added to the References as a new reference.
Perhaps, conclusions could be more elaborate. They are too short from my point of view.
We supplemented some information in the Conclusions, as follows:
In conclusion, the P. Maudiae embryo sac exhibited an Allium type of development. AMOI is an attractive and effective method for the genetic transformation of Paphiopedilum. The fertilization time of P. Maudiae was 50 to 54 DAP, and the best injection time was coincidentally 48 to 53 DAP, which corresponds to the period of fertilization. Thirty-eight protocorms or seedlings were obtained after several rounds of Hyg selection and four randomly selected Hyg-resistant plants were GUS-positive. Semi-quantitative PCR and quantitative real-time PCR analyses revealed the expression of the hpt gene in the leaves of eight Hyg-resistant seedlings. This protocol represents the first effective genetic transformation protocol for any Paphiopedilum species and will allow for expanded molecular breeding programs to introduce useful and interesting genes that can expand its ornamental and horticulturally important characteristics.
Reviewer 2 Report
Please see attached

Author Response
Response to Reviewer 2
Manuscript by Luo et al.,
Title:“Ovule development and in planta transformation of Paphiopedilum Maudiae by Agrobacterium-mediated ovary-injection”
General Comments.
The Manuscript is a beautiful balanced work between pure botany on embyo development and plant transformation of a most interesting plant species, an orchid hybrid, namely Paphiopedilum x maudiae. The manuscript is very well presented, and data supported. I Have a number of minor observations for the authors to consider.
We thank the reviewer for this positive outlook on our manuscript.
My main concern with the molecular work carried out is the utilization of a transformation vector bearing the 35S promoter (lines 447-449, or the authors were suggesting only for the HPT gene?) which is notoriously poorly active in a monocot system, thus makes me wonder if the observed gus activity, which is rather limited and in the veins, is indeed not just an example of endogenous-like activity as opposed to true GUS expression. While I appreciate that the control appeared not showing evidence of gus activity, does not substantiate that the observed colour is a true GUS Expression.
In our experiment, the control did not show GUS gene expression but showed evidence of GUS activity. However, we analyzed the CeFT, GUS and hpt genes both in AMOI at 35, 48, 50, 53 and 150 DAP seeds after AMOI at 48 DAP (Figure 6 f and i), which shows that all three genes were successfully transformed into plants. However, in seedlings, we could not detect the CeFT gene mRNA from AMOI at 30, 35, 42, 43, 45, 48, 50, and 53 DAP, suggesting that a loss or silencing of the heterologous CeFT gene may have occurred during seed germination and growth, or because the Ubi promoter was unable to activate the expression of CeFT in P. Maudiae. Therefore, we used the hpt gene to detect Hyg-resistant seedlings (Figure 6 i and j). We supplemented this information in the Discussion.
General comments.
Name convention for the hybrids is Paphiopedilum x maudiae. The manuscript should be carefully reviewed, and the name rectified.
“Paphiopedilum x …” is a natural hybrid whereas Paphiopedilum Maudiae is an artificial hybrid. The way in which we have written Paphiopedilum Maudiae is correct / accurate.
According to the international systems of measurement, numbers and units must be separated by a space, that also includes percentages and temperature. The manuscript should be carefully review and the name rectified.
This position is debatable, especially the two units that the reviewer refers to. There is considerable inconsistency even in this journal, and we offer two recent examples of plant research from IJMS to support this claim (see % use):
https://www.mdpi.com/1422-0067/21/24/9666
https://www.mdpi.com/1422-0067/21/24/9642
Despite not agreeing entirely with this suggestion, we accepted it and revised the manuscript accordingly.
Full scientific binomial name is to be in italic however genus/genera only are not italicised. Verify throughout the manuscript that it is adhered to the convention.
We respectfully disagree with this suggestion. As concrete examples from this journal, please see two recent examples of plant research from IJMS to support our claim:
https://www.mdpi.com/1422-0067/21/24/9666
https://www.mdpi.com/1422-0067/21/24/9642
Latin names and Latin derived abbreviations, such as et al., are to be italicised”
We accepted this suggestion and revised the manuscript (Line 226).
All percentages should be presented as for accuracy with a single decimal point.
We respectfully disagree with this suggestion. Percentages with two decimal points are accepted in many journals; of note, rounded percentage values with a single decimal point did not show differences among some treatments; therefore, to be consistent with our analyses, which employed two decimal points, we wish to accurately report our data with two decimal points. There are also ample examples of papers (plant science and other) in IJMS published with two decimal points.
Specific Comments
Page 8, line 234. Figure 4 title should not be in bold. The measurements bars are nor clearly readable, but it seems to be 2 mm, which cannot be for all the 5 panels, as the relative size of the seeds is very similar for a, b, c and d but very different from e.
Figure 4 has been revised accordingly.
Line 238 “The percentage of seeds that were not injected (60.61%) and that stained positive for TTC was significantly higher than that of injected fruits,…” it is unclear whether the authors are talking about seeds or fruits. I believe that the authors injected fruits and not seeds, therefore this should be clarified.
The sentence was modified to “The percentage of seeds that stained positive for TTC from the non-injected fruits was 60.61%, which was significantly higher than that of injected fruits at all stages (DAPs)”.
Line 249. “The seeds of non-injected fruits by AMOI were not stained GUS-positive…” did the authors mean to say … The seeds of non-injected fruits by AMOI did not stain GUS-positive…?” currently it is not clear.
We accepted this suggestion and revised the manuscript.
Figure 5. reduce the decimal point to 0 (from 0.00 to 0). Y axe label should be flipped around and changed to: “PLBs Survival Rate (%). X axe label should be Hyg concentration (mg/L). Add in the figure legend the full acronym for PLBs.
Figure 5 has been revised accordingly.
Materials and Methods. A clear presentation of the vector and its promoters is required
The map of the T-DNA was provided in Supplemental information 2.
Line 448 GusA is the Beta glucuronidase enzyme and not the Beta glucose enzyme gene. To be correct is gus expression and the gene is the uidA gene.
We accepted this suggestion and revised the manuscript.
Reviewer 3 Report
The manuscript “Ovule development and in planta transformation of Paphiopedilum Maudiae by Agrobacterium-mediated ovary-injection” presents innovative method of Agrobacterium-mediated plant transformation, particularly useful for resistant species, like orchids. Although the method is not absolutely new, it is still not routine, and several factors affecting its effectiveness need to be determined. Here, Authors investigated the effect of the stage of embryo sac development on transformation frequency. Authors conducted large-scale and meticulous experiments. Especially quality of CLSM analysis is worth to emphasise. Also the design of other analyses and methodology are adequate in general. However, before possible paper publication, some issues need to be adjusted or explained.
Major points
The section “Embryogenic development” is too long and somewhat hard to read. Authors should shorten it since general course of embryo sac development, also in orchids, is known. Hence only distinguishing features of that process in Paphiopedilum Maudiae could be described with details. Moreover, the whole process is very well showed in the Figure 3 and summarised in the Table 1.
Authors calculated the transformation frequency as the ratio of Hyg-resistant plantlets to germinated seeds. However, this parameter could be calculated as the ratio of Hyg-resistant plantlets to all sown seeds – since all seeds (regardless vital or not) are the result of fertilization and concomitant AMOI. Hence, either the presented method of calculation of transformation frequency should be explained/justified or this parameter recalculated. Additionally – statistical analysis of this parameter could be performed to indicate if observed values differed significantly.
Authors analysed the expression of hpt gene and activity of GUS. What about CeFT, which was also introduced via AMOI?
The description (section 4.9) and depiction (Figure 6i) of semi-quantitative PCR of hpt gene is not complete. The quantities of dilutions of primarily obtained cDNA, used as the template for subsequent PCR, should be provided and results for consecutive samples presented (Fig. 6i).
Authors indicated possible reasons of lowered transformation frequency of ovules injected 43-45 DAP. However, the stage of embryo sac development (here four-nucleate and ongoing cell divisions) could be also considered. If Authors agree – they are encouraged to try to indicate possible mechanism of insufficient interaction with Agrobacterium or T-DNA transfer/integration. If they disagree – please response and explain (not in the manuscript).
The description of the vector used for transformation is not informative. It is unintelligible which gene is placed under control of CaMV promoter and NOS terminator – hpt only or GUsA and CeFT too. Although relevant paper is cited, optimally, the map of the vector or T-DNA should be provided.
Minor points
Throughout the whole text – please change “HPT” to “hpt” when it regards the gene. Usually gene names are written lowercase and italic, while their protein products – uppercase and normal font.
Abstract, line 31. Change to “plants were GUS-positive”.
Line 101. Correct to “gametophyte”.
Line 229-232. Here slightly stained seeds are described as with incomplete or no embryo (Fig. 3c) and these with aborted seeds as unstained. However, in the figure caption, the embryo in the panel Fig. 3c is described as aborted. Please clarify.
Line 238. Re-write this sentence as follows: “The percentage of seeds that were not injected and that stained positive for TTC (60.61%)…”. It is somewhat confusing in its current form.
Line 301-302. Re-write this sentence as follows: “PCR analysis and semi-quantitative PCR were used to determine presence of hpt gene and then its expression in (…)”. PCR by itself is not a method for gene expression analysis.
Figure 6 caption. A) captions of panels “f” and “g” are swapped; B) line 315. Correct “lines” to “lanes”.
Figure 7 caption. Change “a: flower of P. Maudiae” to simply “flower” or “flower at the stage suitable for AMOI”. In its current form it is awkward repetition.
Line 448. Correct “β-glucose acid enzyme” to “β-glucuronidase”.
Line 449. Correct “CAMV” to “CaMV”.
Line 490. Currently Sigma Aldrich is a part of Merck group. Change to “Merck-Sigma”.
Line 491. Correct “CW” to “CM”.
Line 513 Why not to use the systematic name of the compound (yellow prussiate of potash), analogously to potassium ferrocyanide?
Line 531. “Agrobacterium” in italic font.
Section 4.8 and 4.9. Use terms “Forward” and “Reverse” for primers instead “left, right”.
Author Response
Response to Reviewer 3
Comments and Suggestions for Authors
The manuscript “Ovule development and in planta transformation of Paphiopedilum Maudiae by Agrobacterium-mediated ovary-injection” presents innovative method of Agrobacterium-mediated plant transformation, particularly useful for resistant species, like orchids. Although the method is not absolutely new, it is still not routine, and several factors affecting its effectiveness need to be determined. Here, Authors investigated the effect of the stage of embryo sac development on transformation frequency. Authors conducted large-scale and meticulous experiments. Especially quality of CLSM analysis is worth to emphasise. Also the design of other analyses and methodology are adequate in general. However, before possible paper publication, some issues need to be adjusted or explained.
Major points
The section “Embryogenic development” is too long and somewhat hard to read. Authors should shorten it since general course of embryo sac development, also in orchids, is known. Hence only distinguishing features of that process in Paphiopedilum Maudiae could be described with details. Moreover, the whole process is very well showed in the Figure 3 and summarised in the Table 1.
We trimmed several sentences and reduced the overall text by about a fifth of the original length.
Authors calculated the transformation frequency as the ratio of Hyg-resistant plantlets to germinated seeds. However, this parameter could be calculated as the ratio of Hyg-resistant plantlets to all sown seeds – since all seeds (regardless vital or not) are the result of fertilization and concomitant AMOI. Hence, either the presented method of calculation of transformation frequency should be explained/justified or this parameter recalculated. Additionally – statistical analysis of this parameter could be performed to indicate if observed values differed significantly.
We accepted this suggestion and revised the manuscript. The percentage of resistant seedlings was calculated as the ratio of Hyg-resistant plantlets relative to all sown seeds. Statistical analysis was also performed.
Authors analysed the expression of hpt gene and activity of GUS. What about CeFT, which was also introduced via AMOI?
We analysed the CeFT, GUS and hpt genes both in AMOI at 35, 48, 50, 53 and 150 DAP seeds after AMOI at 48 DAP (Figure 6 f and i), which shows that all three genes were successfully transformed into plants, but in seedlings we could not detect the CeFT gene mRNA from AMOI at 30, 35, 42, 43, 45, 48, 50, and 53 DAP, suggesting that the loss or silencing of the heterologous CeFT gene may have occurred during seed germination and growth, or because the Ubi promoter was unable to activate the expression of CeFT in P. Maudiae. Therefore, we used the hpt gene to detect Hyg-resistant seedlings (Figure 6 i and j). We supplemented this information in the Discussion.
The description (section 4.9) and depiction (Figure 6i) of semi-quantitative PCR of hpt gene is not complete. The quantities of dilutions of primarily obtained cDNA, used as the template for subsequent PCR, should be provided and results for consecutive samples presented (Fig. 6i).
The consecutive samples were provided in Supplemental information 3.
Authors indicated possible reasons of lowered transformation frequency of ovules injected 43-45 DAP. However, the stage of embryo sac development (here four-nucleate and ongoing cell divisions) could be also considered. If Authors agree – they are encouraged to try to indicate possible mechanism of insufficient interaction with Agrobacterium or T-DNA transfer/integration. If they disagree – please response and explain (not in the manuscript).
The possible reasons of lower transformation frequency of all sown seeds form ovules injected 43-48 DAP are that embryo were easily damaged by AMOI during the four- to eight-nucleate embryo sac stages.
The description of the vector used for transformation is not informative. It is unintelligible which gene is placed under control of CaMV promoter and NOS terminator – hpt only or GUsA and CeFT too. Although relevant paper is cited, optimally, the map of the vector or T-DNA should be provided.
The map of the T-DNA was provided in Supplemental information 2.
Minor points
Throughout the whole text – please change “HPT” to “hpt” when it regards the gene. Usually gene names are written lowercase and italic, while their protein products – uppercase and normal font.
We accepted this suggestion and revised the manuscript.
Abstract, line 31. Change to “plants were GUS-positive”.
We accepted this suggestion and revised the manuscript.
.
Line 101. Correct to “gametophyte”.
We accepted this suggestion and revised the manuscript.
Line 229-232. Here slightly stained seeds are described as with incomplete or no embryo (Fig. 4c) and these with aborted seeds as unstained. However, in the figure caption, the embryo in the panel FiHabg. 4c is described as aborted. Please clarify.
We revised the expression in the text and the caption of Fig. 4 in the manuscript.
Line 238. Re-write this sentence as follows: “The percentage of seeds that were not injected and that stained positive for TTC (60.61%)…”. It is somewhat confusing in its current form.
According to the opinion of the reviewer 2 and reviewer 3, the sentence was modified as “The percentage of seeds stained positive for TTC from the fruits that were not injected was 60.61%, which was significantly higher than that of injected fruits at all stages (DAPs)”.
Line 301-302. Re-write this sentence as follows: “PCR analysis and semi-quantitative PCR were used to determine presence of hpt gene and then its expression in (…)”. PCR by itself is not a method for gene expression analysis.
We accepted this suggestion and revised the manuscript.
Figure 6 caption. A) captions of panels “f” and “g” are swapped; B) line 315. Correct “lines” to “lanes”.
We changed captions “f” to “g” and “g” to “f” in Fig. 6.
Figure 7 caption. Change “a: flower of P. Maudiae” to simply “flower” or “flower at the stage suitable for AMOI”. In its current form it is awkward repetition.
We accepted this suggestion and changed to “flower” in the manuscript.
Line 448. Correct “β-glucose acid enzyme” to “β-glucuronidase”.
We accepted this suggestion and revised the manuscript.
Line 449. Correct “CAMV” to “CaMV”.
We accepted this suggestion and revised the manuscript.
Line 490. Currently Sigma Aldrich is a part of Merck group. Change to “Merck-Sigma”.
We accepted this suggestion and revised the manuscript.
Line 491. Correct “CW” to “CM”.
We accepted this suggestion and revised the manuscript.
Line 513 Why not to use the systematic name of the compound (yellow prussiate of potash), analogously to potassium ferrocyanide?
“yellow prussiate of potash” was revised to “potassium ferricyanide”.
Line 531. “Agrobacterium” in italic font.
We accepted this suggestion and revised the manuscript.
Section 4.8 and 4.9. Use terms “Forward” and “Reverse” for primers instead “left, right”.
We accepted this suggestion and revised the manuscript.